# Diet Quality, Health, and Wellbeing within the Irish Homeless Sector: A Qualitative Exploration

**DOI:** 10.3390/ijerph192315976

**Published:** 2022-11-30

**Authors:** Divya Ravikumar, Elena Vaughan, Colette Kelly

**Affiliations:** Discipline of Health Promotion, School of Health Sciences, University of Galway, H91 TK33 Galway, Ireland

**Keywords:** homelessness, diet, health inequalities, food security

## Abstract

Financial barriers and limited cooking facilities are major obstacles to healthy dietary practices among the homeless population. Homelessness is currently at crisis point and up-to-date evidence from multiple stakeholders is needed to address dietary inequalities. The aim of this study was to understand dietary practices, barriers to healthy eating within homeless services from multiple perspectives. Twelve service users and five healthcare and social service providers participated in semi-structured interviews. Data were analysed thematically. Four themes were identified which included: lack of control over diet and food supply; sources of food for the homeless population; practical barriers to good nutrition; and the impact of diet on emotional and physical wellbeing. Frequent consumption of energy-dense, nutrient-poor foods was reported. Food insecurity resulted in perceived depressive symptoms and stress. Barriers to healthy diet included financial constraints and a lack of access to cooking and storage facilities. Our study highlights low levels of food skills and healthy eating knowledge among service users and service providers. In order to address diet-related health disparities, health promotion initiatives should be targeted at building healthy public policy in relation to diet and nutrition and developing food skills with members of this population and service providers.

## 1. Introduction

There are 2.1 million people experiencing homelessness across 36 OECD (Organisation for Economic Cooperation and Development) countries [1]. In recent years, there has been an increase in homelessness across one-third of OECD countries, including Ireland, England, the United States, Australia, New Zealand, France, Wales, Scotland, Luxembourg, Latvia, the Netherlands, Iceland, and Chile [1]. In the United States, over 550,000 people are experiencing homelessness based on a single night estimate in January 2019, which is a 3% increase since 2018 [2]. In the UK, the number of households experiencing homelessness in June 2020 was 98,300, a 14% increase since the same month in the previous year [3]. Figures from June 2022, indicate that over 10,000 Irish people are experiencing homelessness, including approximately 3000 children and 1400 families [4]. This is an increase of 65% since June 2016 [4,5].

Those who are homeless rely on social services and donations for food, with local food businesses often donating excess food of low nutritional value to shelters or food banks [6]. In the US, soup kitchens and homeless shelters are a key source of food for people who are homeless [6,7,8], and food banks set up by Feeding America are used by over 46 million people every year [9]. Use of food banks and charitable meal services has risen exponentially in many parts of the UK and Ireland over the last decade [10,11].

There has been a change over time in the metabolic health status of the homeless population from primarily underweight to obese [12], resulting in a chronic state of both obesity and hunger which is referred to as the hunger-obesity or food-security obesity paradox [13,14,15]. In the US, obesity in the homeless population is reported at 39% [12,16,17] and this population has a 3–5 times greater risk of mortality and a higher-than-average burden of cardiovascular mortality and morbidity than the general population [18]. In an Irish study, 90% percent of a homeless sample exhibited abdominal obesity [19], while a German study examining nutritional health in an urban homeless population showed that two thirds of the population suffered from at least one chronic disease [20]. Poor control of hypertension, diabetes and other cardiovascular risk factors were highlighted in studies on the homeless population due to decreased likelihood to present to hospitals, poor adherence to medications and other competing priorities [19,21].

Food insecurity has been linked to increased levels of post-traumatic stress disorder and depressive symptoms [22]. Despite the high prevalence of depression, personality disorders and psychotic illness among both adults and children in the homeless population [23,24], research exploring diet and mental wellbeing in the homeless population is very limited. A UK study found that people who were homeless compared to those who are housed have a higher incidence of depression and anxiety and poorer dietary and nutrient intake [25]. People experiencing homelessness have reported that the routine associated with receiving meals at shelters negatively impacted on their emotional wellbeing and resulted in people seeking out high energy, low nutrient density foods in order to achieve a sense of comfort or freedom [26].

Practical barriers such as local policies in shelters and temporary accommodation have been documented as barriers to good nutrition within the homeless population in the US [27,28]. Examples include the prohibition of perishable foods in shelter bedrooms, early dinner times resulting in the need for non-perishable snacks, a lack of food storage facilities and not being offered fruit and vegetables in dining facilities. In Ireland, limited cooking and food storage facilities were barriers to good nutrition and lead to an increase in the consumption of non-perishable goods [29]. Restricted access and availability of facilities were also the primary challenge facing families who were homeless in relation to the provision of nutritious food [26].

Only two Irish studies focusing on homelessness and diet are currently in the literature. Both studies documented a lack of cooking and storage facilities, low fruit and vegetable consumption, high intake of pre-prepared meals, takeaways and processed food, weight gain and a sense of powerlessness over food choices [26,29]. One study focused on those in emergency and hostel accommodation only and was specific to parents with children [26]. While valuable in terms of type of homelessness experienced [29] the findings lack currency due to the significant increase in homelessness in Ireland in the past 18 years.

The aim of this study is to understand dietary practices, barriers to consuming healthy food and perceived wellbeing in individuals experiencing homelessness.

## 2. Materials and Methods

### 2.1. Researcher Characteristics and Reflexivity

The primary researcher was undertaking a master’s thesis at the time of data collection and is currently undertaking a PhD in this field. The other team members have extensive qualitative research experience. Participants were not previously known to the primary researcher prior to undertaking this study.

Access to service users was facilitated through two homeless service centres in a city in the West of Ireland. Purposive and snowball sampling was used for the purpose of recruitment [30]. Social care service providers and health care providers facilitated access to participants; this approach was successful in previous studies [19,26]. Recruitment ceased when data saturation was reached.

### 2.2. Sampling Strategy

Inclusion criteria for service users were adults over the age of 18 years that were homeless in the city and availing of sheltered and emergency accommodation or sleeping rough, which refers to sleeping outside and in places not designed for living such as empty buildings and cars [31]. The length of time spent in homelessness did not affect recruitment to the study. Inclusion criteria for service providers were adults over the age of 18 years that were providing healthcare and social services to members of the homeless population in the same city. The length of time spent providing services to the homelessness population did not affect participation in the study.

### 2.3. Ethical Issues Pertaining to Human Participants

Ethical approval for this study was granted by the post-graduate ethics committee within the Health Promotion Discipline at the University of Galway. Informed consent was gained from all participants that took part.

### 2.4. Data Collection Methods

Interviews were conducted in private rooms in public venues, including health and homeless service centres.

Interviews lasted on average one hour and were recorded using a digital audio recorder. As a back-up measure, a password protected mobile phone was also used to record information.

### 2.5. Data Collection Instruments and Technologies

Semi-structured interviews were carried out with service users and providers of health and social care services, to gain insight into the dietary habits and barriers to accessing healthy food within the context of homelessness. A topic guide was used to prevent the interview from veering away from the areas of interest [32]. Examples of questions asked to service users included: (1) Give me an idea about what a typical day would be like in relation to your diet; (2) Where do you obtain most of your meals? (3) Can you describe any cooking facilities you use?; (4) Tell me about the impact that eating in this way is having on your physical/emotional wellbeing. Examples of questions asked to service providers include: (1) How would you describe the diet of people that are experiencing homelessness?; (2) From your experience, what are the main practical barriers to eating healthy food among this population?; (3) Do you have any insight into the impact that eating in this way is having in terms of physical/emotional wellbeing in this population?; (4) Do you think that there are improvements that need to be made to services that provide food for people experiencing homelessness?

### 2.6. Data Processing

All recorded data were transcribed manually by one researcher, and NVivo 12 [33], was used to manage the data. All audio files were stored on a password-protected USB and secured in a supervisor’s files in University of Galway. All audio recordings were deleted once the transcription was completed and all transcriptions had been checked by the researcher. Consent forms were scanned and stored on an encrypted USB and hard copies of the data were destroyed after each interview. Soft copies of data were stored on the University approved cloud storage facility, Microsoft OneDrive [34], which is password protected. Any data stored on Microsoft OneDrive had participant names removed prior to uploading the data.

### 2.7. Data Analysis

Reflections were written following each interview in order to document any challenges or issues that arose during the interview. These notes were reviewed during transcription, coding, and data analyses. Thematic analysis was conducted using a six-step process to data analysis [35]. Data analysis, coding and identification of themes was initially carried out by one researcher. All codes and potential themes were discussed with the project supervisor prior to agreement on final themes. Data from service users and service providers were initially coded separately. However, analyses revealed overlapping data and themes, so the data have been presented together with differences highlighted within themes.

## 3. Results

Seventeen interviews were conducted comprising twelve service-user interviews (9 men and 3 women) and five service-provider interviews (3 men and 2 women). The length of time service users had spent in homeless services ranged from 2 months to 12 years. Of the 12 service users, 6 were residing in emergency accommodation and 6 were residing in hostel accommodation. Service providers included a chef in a homeless day service, a social service provider in a homeless hostel, a social service provider that works with families in emergency accommodation and a nurse and general practitioner, both working primarily with the homeless population. The length of time service providers had worked in homeless services ranged from 8 months to 23 years.

The data analysis process yielded the following themes from service user and service provider interviews: (1) Lack of control over diet and food supply; (2) Sources of food for the homeless population; (3); Practical barriers to good nutrition; (4) The impact of diet on emotional and physical wellbeing. A sample of quotations to represent each theme can be seen in Appendix A.

### 3.1. Theme 1: Lack of Control over Diet and Food Supply

This theme captures the inability of service users to make choices about their diet, prepare food and exert control over their diet, as well as the issues faced by service providers in exerting control over food supply to homeless services. In emergency accommodation, a loss of control was highlighted in relation to being able to provide for children. This was expressed by mothers in the study, both of whom struggled with their inability to provide for their children. Both mothers stated that this is due to limited cooking facilities and described the challenges of trying to explain this to their children and the consequent frustration they felt: “I can’t cook it for you… but she (daughter) just, she didn’t understand” [Woman, service user].

Although some service users within hostel accommodation were happy to have food provided for them, others felt that lack of control was an issue. The inability to cook meals was also stressed here. A service user supplemented his diet in order to be in command of his own food choices because “the food… becomes monotonous” [Man, service user], in the hostel. Another service user highlighted his desire to cook: “I’d love the chance to do that myself… they don’t really have anything other than boiling water to offer”.

This issue could also affect families and was highlighted as a “main topic of conversation” amongst service users by the social service provider for families in emergency accommodation. She described the difficulties families can face on a special occasion such as Christmas: “Even for Christmas day, you know, to even be able to cook dinners, families would say, ‘We sat in this hotel room on Christmas day and we didn’t even have a dinner’” [Woman, social service provider].

Controlling food donations was described as difficult by service providers, despite health and safety policies in place. The policy in services is to refuse food donations due to the possibility of contamination but a service provider highlighted that this can put service providers in an awkward position when members of the general public try to donate food. Homeless services often rely on the general public as a source of funding and this can make service providers reluctant to deny these food donations out of fear of offending people. In practice, this food is consumed by service users or is accepted by service providers and then immediately discarded. Supermarkets also donate food through services such as Food Cloud [36], including vegetables, sliced meats, bags of potatoes or whatever they have “leftover or on short shelf life.” These donations were perceived as useful to chefs as they can be included in the cooking process.

### 3.2. Theme 2: Sources of Food for the Homeless Population

Food was provided on site by both emergency and hostel accommodations. However, the number of meals and type of food received varied between the two types of accommodation. Participants staying in hostel accommodation had the option of receiving all meals on site and an emphasis was placed on ensuring food was freshly prepared:

“There’s always going to be veg (vegetables)… and it’s all freshly cooked. Our soups are all homemade. You know nothing comes out of a can… The lads (men) do get dessert every day”[Man, social service provider]

He also highlighted a high prevalence of sugary foods, “the cakes and the biscuits can slip in as well”. Some service users may also eat two dinners as leftover food goes “back out again at 8 o’clock”. Many service users confirmed that food was freshly prepared, plentiful and easily available within hostel accommodation.

All participants residing in emergency accommodation were provided with breakfast in their accommodation which usually consisted of “a full Irish (fried bacon, sausages, black and white puddings, eggs, bread, cooked potato/potato bread, mushrooms, tomatoes [37]), “cereals… yoghurts… fruit” (Woman, service provider). For the rest of the day, service users had to provide for themselves, and no food was available on-site following breakfast. This resulted in frequent consumption of “cheap takeaways”, “cereals, pot noodles” or “anything convenient”. This was affirmed by service users who reported frequent consumption of energy-dense, nutrient-poor sources of food and a lack of vegetables: 

“The closest I really get to vegetables now is… peppers on a pizza or something”[Man, service user]

When sleeping rough food choice was dominated by affordability or charity. Members of the homeless population with experience of sleeping rough, had an awareness of the cost of food within their area and how to make the most of the money they received: “The cheapest thing (is)… €4.50… that’s what would feed me in a day…that would get you a chicken roll, wedges, and a drink. You can’t really beat that anywhere else” [Man, service user]. When sleeping rough, an awareness of how to obtain free food was also noted:

“People who are homeless will watch people when they’re drunk. Students—they’ll buy a pizza, and they take one piece, and they leave it on the wall, so people live off that”[Man, service user]

### 3.3. Theme 3: Practical Barriers to Good Nutrition

A lack of facilities to cook and prepare food was described by many. Cooking facilities ranged from having no cooking or storage facilities to having a kettle, microwave, and a communal fridge. As mentioned previously, all meals are provided in hostel accommodation. However, within emergency accommodation, only one service user in the study had access to a cooker. Attempting to prepare food without cooking or storage facilities proved very difficult:

“I went in and bought a half pan of bread, ham and cheese, coleslaw…butter and I went home that night and I think I must have had about six sandwiches but then I had to bin… whatever was left”[Man, service user]

This service user went on to comment that this was also difficult because food had to be consumed within one’s own room, “it doesn’t work. And then you have butter now on your bed”.

Limited or no cooking facilities resulted in the need to purchase two meals a day which proved very challenging for service users: “Eating out is expensive. You know I find, say dinners, you know, I can get a dinner for maybe €10, it could be €12, €13 maybe some days” [Man, service user].

The lack of evening meals from emergency accommodation services was highlighted as a problem by two service users that were women. Both participants expressed that they would be willing to pay some money towards this to ensure that they consumed one “warm”, “home-made” meal during the day. This was also emphasised as a particular issue for service users that are elderly or have health conditions that may make it difficult to travel outside of their accommodation to obtain food.

Difficulty in relation to food provision for young children in emergency accommodation was reported during this study. An inability to provide age-appropriate food for a toddler was mentioned due to a lack of facilities for weaning children onto solids: “You know the baby jars… you can’t keep giving them. I’d like to give her… proper (food)… be able to cook it and give it to her… vegetables and meat… she should be having that at her age” [Woman, service user].

This participant also had difficulty obtaining breakfast even though breakfast is provided in her accommodation. This is due to the short period for which breakfast is served and the difficulty readying her children for school in a confined space. “Most days I do skip breakfast… like if we had our own home, I would have the time like because they’d be sitting down at a proper table, eating their own breakfast. I’d be able to sit down with them. Because we’re in a bedroom, they’re all up on top of each other” [Woman, service user].

From a service provider perspective only, several participants mentioned a lack of knowledge around healthy food choices and a need for nutrition training. Staying in hostel accommodation on a long-term basis was highlighted as a way to “de-skill you” and suggestions were made for tailored education resources such as a booklet with “cheap, healthy food” and “simple little recipes” for the homeless population. This was mentioned as something that could be an addition to the “move-on plan” that is put together when service users are planning on moving into private accommodation. Nutrition training for service providers was also deemed necessary as it is not core to social care/services training but was recognised as a need in this study: “I think it’s something that our organisation has just acknowledged in the last few months, this is why I started, is that people have came into homeless services with the idea that you’re in private emergency accommodations for four to five months and this has now extended to nearly two years… and we’ve just realised this isn’t acceptable so we need to kind of encourage healthy eating and maybe, yeah, training” [Woman, social service provider].

A healthcare provider also emphasised this need, describing social service providers as “the gatekeepers, in that they control the food that comes into the house” [Man, healthcare provider]. He went on to say that “if the staff were educated, I think that would filter through”. Inadequate resources such as recipe booklets were also reported as an issue. However, it was also highlighted that these booklets would need to be tailored in order to ensure they accounted for varying literacy levels within the homeless population.

### 3.4. Theme 4: The Impact of Diet on Emotional and Physical Wellbeing

In relation to physical wellbeing, the negative effect on oral health in children since moving into homeless services was described: “The most obvious effect to him would be the cavities, you know… when you look at his dental records from before he went in (to homeless services), we had no problems, there was zero issues” [Woman, service user, mother]. An impact on weight was described for both children and adults. Mothers expressed concern that their children had gained considerable weight (2–2.5 stone/12.7–15.9 kg approximately) during the two years since entering homeless services. Healthcare practitioners also noted “high levels of obesity, high cholesterol levels, raised blood sugar” in their work with the adult homeless population [Woman, healthcare provider]. The positive impact of weight gain when moving from sleeping rough to homeless services was noted: “It’s remarkable the difference because I put on about ten kilos in the space of a month, when I came in here… I got very light, very skinny out there when I was homeless [sleeping rough]” [Man, service user].

The perceived impact of food provided within homeless services on emotional and mental health varied amongst service users. Several participants in hostel accommodation described the benefits of homeless services on their emotional health in relation to food security: “Without having to worry about food as much as before (like), it kind of takes a lot of stress off your mind, let’s you focus on other things. (Like) emotionally I feel a lot better because I’m not stressing about my next meal” [Man, service user].

However, another participant in this type of accommodation experienced depression when he struggled to obtain foods suitable for a medical condition that required a puréed diet: “I lost loads of weight and I was depressed constantly. I was suicidal” [Man, service user].

A vegetarian participant also highlighted that he skipped meals due to the cost of eating out, “I never have three meals in a day… you know, which I would have before”. He found it particularly challenging to cater to the vegetarian diet when one must buy meals in inexpensive food establishments, “I’m a bit restricted, being vegetarian cause most of what’s on the menu is meat or fish or any of that” [Man, service user]. This feeling was compounded by worries about perceived judgment and stigma that may result from visiting the same establishments on a regular basis: “I don’t want to be going into the same place like three times a week cause… I don’t want them to be kinda looking at me thinking, look at him coming ‘round here three days a week… He must eat junk food every day, you know” [Man, service user].

Healthcare providers also mentioned comfort eating or emotional eating due to stress or boredom as perceived issues within this population: “I think a lot of them turn to food probably in times of stress” [Man, healthcare provider]. An improvement in mood once families move into self-catered accommodation (accommodation with cooking and storage facilities) was also reported “Cause people who have went into self-catering… even the children… you can see the children are so much happier” [Woman, healthcare provider].

## 4. Discussion

Homelessness can be classified in a number of ways due to the various lifestyles that exist under this term. Homelessness includes those who are “transitionally homeless”, i.e., living in emergency accommodation such as a bed and breakfast, hotel, hostel or sofa surfing with friends and family; those who are “episodically homeless” and may be in and out of homelessness and suffer from residential instability; and those who are “chronically homeless” and are more likely to illegally occupy buildings (squatting), sleep rough or use shelters [38,39,40]. The various types of homelessness make it difficult to compare data from people who are classified differently. This, along with the transient nature of this population [6], can make conducting research on this population very challenging.

The findings from this study have shown that dietary habits in people sleeping rough are primarily focused on cheap takeaway food such as pizza, chips and burgers as well as high-sugar drinks. This cohort often relied on charity from members of the public to obtain fast food. Charitable organisations also provided food such as soup or sandwiches for rough sleepers.

All participants in emergency accommodation in this study had access to a cold or hot breakfast, reflecting previous findings [26], however, participants in emergency accommodation in this study did not have access to a second meal during the day. High consumption of fast food and takeaway meals and convenience foods in this cohort has also been previously documented [26,29]. Low vegetable consumption in this group has been found previously in Ireland [26]; Germany [20]; and the UK [41].

In contrast, meals in hostel accommodation were structured and scheduled, with multiple options and vegetables at every meal. Sugary food such as cakes and biscuits were readily available, and dessert was given to all residents after dinner. These sugary foods included food donations from members of the community. A study on dietary habits in Boston food shelters also reported the donation of high-sugar foods such as left-over dessert and pastries [6].

The financial cost of food was a major barrier to obtaining healthy food options among members of this cohort. Participants highlighted that buying meals from food outlets was expensive and finding healthy food was difficult. The provision of only one on-site meal in emergency accommodation resulted in service users being forced to spend money on food from outside establishments despite the significant financial burden. Previous research has also found that eating healthy food is more expensive than more nutrient-dense foods and an inverse relationship exists between energy density and energy cost [42]. Financial barriers related to the cost of sourcing healthy food in food outlets is not surprising with other studies reporting “difficulties in reconciling their tight budgets with the principles of healthy eating” [29]. Studies from the US have also found that families and children experiencing homelessness had limited choice of food due to the inflated costs, poor variety and poor quality of food [43,44].

All service users in emergency accommodation that took part in this study raised the issue of access to cooking and storage facilities which limited food choices and resulted in high consumption of takeaway and convenience food. This was a particular issue for parents in emergency accommodation, who expressed frustration over their inability to cater to their children’s most basic needs and desires in relation to food. Poor weaning practices, as reported by Share & Hennessy [26], were also highlighted as an issue for parents in this study due to inadequate kitchen facilities.

A sense of powerlessness over one’s dietary choices was expressed, with service users in this study reporting that they were often unable to eat the food that they or their children desired. This has been previously reported in Irish studies [26,29]. Similarly, research in the US has shown that policies prohibiting the storage of perishable food in sheltered accommodation, a lack of storage facilities and a limited availability of fruit and vegetables act as a barrier to the consumption of healthy food [27].

The findings suggest diet may have an impact on both physical and emotional health. Weight changes related to diet were commonly reported; participants in emergency accommodation and hostel accommodation reported an increase in weight upon entering services which was viewed both positively and negatively. High levels of obesity were reported by service providers also. Previous US research in this area has shown high levels of overweight and obesity to be as high as 58.8% [12,45,46]. Irish research has shown abdominal obesity levels to be as high as 90% [19]. Increases in weight were also documented in two Dublin-based studies on diet in the homeless population [26,29].

Other physical health concerns related to diet include raised cholesterol and raised blood sugar. Previous US research shows that people experiencing homelessness have a higher-than-average burden of cardiovascular mortality and morbidity [12,16,18].

Emotional health was shown to be impacted by diet in this study. Low mood and high levels of stress were reported by service users and service providers. Stress in relation to food security was raised as an issue with both service users and service providers and this corresponds with previous studies [22]. Emotional eating was also highlighted as a concern in relation to emotional health in this study. This impact on emotional wellbeing was seen in another Irish homeless cohort where the routine nature of meals provided in shelters resulted in people consuming cheap energy-dense, nutrient-poor in search of comfort or a sense of freedom [26].

Service providers in this study reported food knowledge and skills as a barrier to the consumption of healthy food within the homeless population. This was raised as an issue for people staying in homeless hostels, in particular, as all meals are catered in this service. A lack of food skills in those that were previously in the care of social services was apparent and the need to improve skills prior to transitioning from hostel to private accommodation was highlighted. Similar to previous research, this suggests that being in care may reduce opportunities to learn to prepare, cook and store food [29]. This may also relate to food literacy and a person’s ability to source food and budget for food shopping [47].

Service provider knowledge may also be a determinant of diet in the homeless population. This study highlighted a need for education and training on nutrition and food skills for service providers that work with the homeless population. Currently, advice on healthy eating and food skills is primarily given through conversations between service users and providers, which further strengthens the need for service users to be educated on healthy eating. This study stressed that service providers act as gatekeepers of food in homeless services and this is another reason that they should be the target group for nutrition training.

However, initiatives to provide food skills to people experiencing homelessness may not be a useful endeavour unless people are able to turn knowledge into practice. Structural barriers that limit self-catering opportunities have been shown to hinder efforts to improve the nutritional knowledge of both adults and young people experiencing homelessness, with clear negative implications for health and development [48,49]. It is evident that increasing the food knowledge and skills of individuals that experience homelessness may be of little benefit without significant progression towards goals and recommendations outlined in national and local policies governing housing and accommodation provision for the homeless [26,50,51,52]. Without these changes, people residing in homeless accommodation will be powerless over their food choices.

### Strengths and Limitations

This study adds to the evidence base by investigating dietary habits and practical barriers to good nutrition among service users and providers, since the steep increase in the number of people experiencing homelessness in Ireland. A wide range of experiences were captured in this piece of research. Service users with experiences of sleeping rough, sofa surfing, emergency accommodation and hostel accommodation were included in this study. A wide cross-section of service providers also participated in this study, including healthcare practitioners, social service providers and a chef. This diverse group of participants provided rich data that approached the research questions from multiple perspectives.

As the sample for this study was self-selecting, this leaves the potential for selection bias. Data saturation was used to determine the number of participants recruited for this study. The assumption that further interviews could not result in further insights is inherently biased. The lack of gender balance amongst participants was a limitation in this study also. However, this is reflective of the gender balance within the homeless population in Ireland [6]. Participants were predominantly white, with only one participant being from an ethnic minority. Future research should aim to capture data on the food experiences of people from a diverse range of ethnic and cultural backgrounds within homeless services. This is particularly important as these services may not cater for culture-specific diets. The sample was also limited to service users in Ireland and results may not be generalisable to other settings. All participants in this study were engaged in homeless services. Although data were collected on past experiences of service users prior to involvement in services, this study does not reflect the experiences of those that are homeless and are not currently engaged with services.

## 5. Conclusions

The dietary habits of those who are homeless remain poor. No advances in access to or in the provision of healthy food to the homeless population were identified. The barriers identified in previous studies remain, including financial constraints and a lack of access to cooking facilities. Our study highlights low levels of food skills and healthy eating knowledge among service users and service providers. Food insecurity was also linked to perceived depressive symptoms and stress.

This study was conducted with both parents and single people experiencing homelessness as well as service providers that worked with these groups. It adds to the literature by updating the evidence in light of a growing homeless population. Given the increase in the number of people experiencing homelessness and the clear evidence of poor diet, along with the links between diet and physical and emotional health, the impetus for political action should be apparent. However, based on our findings, little to no improvement has been made to the nutritional quality of food and access to cooking and storage facilities within homeless services.

To address diet-related health disparities in this population, health promotion actions must be taken at multiple societal levels. The type of accommodation provided to people experiencing homelessness, with the exception of self-catered accommodation, is not conducive to a healthy diet. This is a wider political issue that stems from ineffective housing policies for the homeless population. This study highlights that health promotion initiatives should be targeted at building healthy public policy in relation to housing, diet and nutrition and developing food skills with members of this population and service providers.

## Data Availability

Data produced during this study are available from the corresponding author upon request.

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
