# Peer review of "Diet Quality, Health, and Wellbeing within the Irish Homeless Sector: A Qualitative Exploration"

_ijerph, 2022, doi:10.3390/ijerph192315976_

Round 1

Reviewer 1 Report

Title should reflect the qualitative nature of the content.

Abstract is well written.

Introduction articulated the past literature and research gap.

Ethics approval number must be provided. 

Any justification for the number of subjects interviewed? Was data saturation achieved with the number interviewed?

Results & Discussion are well described. 

Author Response

Dear Dr. Lemacks,

Many thanks to you and the reviewers for their considered comments on our paper: ‘Diet Quality, Health and Wellbeing within the Irish Homeless Sector: A Qualitative Exploration’ to be considered for publication in the International Journal of Environmental Research and Public Health special Issue on "Addressing the Health Needs of Vulnerable Populations."

We would like to take this opportunity to thank you and the reviewers of our manuscript. The feedback provided was very valuable and we believe the overall quality of the manuscript has increased. Our response to the reviewer comments is highlighted below in red and has been addressed with tracked changes in the main article.

We confirm that neither the manuscript nor any parts of its content are currently under consideration or published in another journal. All authors have approved the manuscript and agree with its submission to the International Journal of Environmental Research and Public Health special Issue on "Addressing the Health Needs of Vulnerable Populations."

Thank you very much for your consideration of this manuscript

Yours sincerely,

Divya Ravikumar

Discipline of Health Promotion

University of Galway

Reviewer 1: Comments

Title should reflect the qualitative nature of the content.

Thank you for this feedback. In order to address this, the title has been changed to ‘Diet Quality, Health and Wellbeing within the Irish Homeless Sector: A Qualitative Exploration.’

Ethics approval number must be provided.

The ethical approval number: DR-04-19 has been included in the manuscript document and a signed letter to confirm ethical approval has been attached.

Any justification for the number of subjects interviewed? Was data saturation achieved with the number interviewed?

Purposive and snowball sampling was used for the purpose of recruitment (Painkas etc Al., 2015). Social care service providers and health care providers facilitated access to participants; this approach was successful in previous studies (Scott et al., 2013; Share & Hennessy, 2017). Recruitment ceased when data saturation was reached.

Reviewer 2 Report

The aim of this study was to describe dietary practices, including barriers to healthy eating, and health perceptions among individuals experiencing homelessness in Ireland. Given the increase in homelessness across the globe in the past years and older publications around this topic area, this paper will add meaningful, current information to this research area.

Introduction

Line 25: Define OECD the first time it is used.

Lines 37, 45, 66: Suggest using United States instead of America.

Line 48: Recommend changing the percentage to 90%.

Line 75: Recommend changing “veg” to vegetable.

Materials and Methods

Line 110: Was a digital recorder and a mobile phone used as a security measure, to ensure interviewers were recorded? If so, that could be helpful to point this out for the reader.

Lines 126-129: The “Units of study” section does not add anything meaningful to the Methods section, so recommend omitting.

Line 130: Ensure NVivo is properly cited.

Line 131: The phrase “audio tapes” makes it sound like you used a manual recording machine, but you digitally recorded the interviews based on line 110. Thus, more appropriate would be to call them audio files. Was the mobile phone password protected? Were the digital recorders stored in a locked filing cabinet? Adding a little more information about this would be helpful.

Line 132-133: I think the sentence “Data collected was only discussed anonymously with the project lead” seems out of place. I think you could omit this sentence, with the efforts used by the team to secure the data still being sufficiently described (with the additions of my comments for line 131).

Line 137: Ensure Microsoft OneDrive is properly cited.

Lines 145-148: Clarify the discussion of the project with the supervisor. Did the supervisor read through all the transcripts to help decrease potential bias with only one researcher coding the data and developing themes? Also, the information seems to be written out of order – describe it in the order it occurred. For example, I am assuming the reflections were written first, then the data were coded, and then codes/themes and subthemes (however, no subthemes were identified in the Results so suggest taking out subthemes and just stating themes) were discussed with the supervisor. Some of the information in this section seems repetitive as well, so this needs to be modified.

Line 146: I am not sure what the author means by “no challenges arose during this process” – suggest omitting this statement.

Results

Lines 151-152: How did researchers determine gender as men and women? Were they asked in the interviews, or did they complete a demographics questionnaire? This information should be described in the Methods section.

Lines 152-159: Recommend grouping all the information about service users together (length of time and residency location) and service providers together (length of time and job description).

Theme 1

Be consistent in clarifying if the ideas are based on service users’ or service providers’ perspectives.

Lines 167-172: These sentences seem to overlap. Recommend combining these ideas. With the quote, it could be helpful to define what “it” is referring to in brackets.

Line 174: How did service users’ sense of control vary?

Lines 183-193: This paragraph is somewhat hard to follow and seems contradictory in some locations. Some food donations are thrown away because of safety concerns, yet the service providers rely on supermarket food donations. Does this vary depending on the source of donations? This needs to be better clarified.

Theme 2

Line 195: I think accommodation should be in plural form (accommodations).

Lines 199-201: This quote is talking about the types of foods and preparation methods at hostel accommodations but from the sentence prior to this quote, it is supposed to illustrate receiving meals and snacks on site. The quote does not illustrate this.

Lines 204-205: How are researchers defining “high quality” foods? Was this the wording given by the service users? High prevalence of sugary foods and encouraging overeating (two dinners), from a nutritional standpoint, might not be considered high quality. Perhaps the perception was that the food was tasty or highly preferred by service users?

Line 216: Recommend adding a comma after “when sleeping rough” – also, I am assuming “sleeping rough” means the individual is a non-housed individual who is experiencing homelessness. If this is correct, it could be helpful to describe this earlier, the first time you use this term, for those who are not familiar with this term.

Lines 222-225: Add quotation marks to this statement. Also, it’s a little unclear if the soup is being given for free based on the first part that says soup is for sale.

Theme 3

Recommend moving this theme to the last one because all of the other themes describe situations that could lead to the outcomes mentioned in this theme.

Lines 228-230: Was this mother talking about changes to cavities since being homeless? The quote is not super clear about this.

Line 232: It seems surprising that mothers would report such exact weight gain ranges. Did they really express ranges from 12.7 to 15.9 kg?

Line 241-243: Add quotation marks.

Lines 247-252: This section is somewhat hard to follow with the information provided prior to it. The sentence prior to this is talking about not having medical foods led to depression, whereas the next sentence is talking about using food to address stress or boredom, and then the last sentence talks about improved mood. Although these are related to mental and/or emotional health, they seem somewhat disconnected. Recommend revising this section for clarity.

Line 250: What is “self-catered accommodation”? I’m not familiar with this term.

Theme 4

Line 254-260: Is the lack of cooking and storage facilities in the emergency accommodation or hostel accommodation or both? It would be helpful to clarify this.

Line 259-260: Add quotation marks.

Line 260-265: How does this experience in spending more money on eating out differ from the lines 254-260? It seems like this is a further example of how limited cooking facilities negatively affected food purchased and how much money was spent on food expenditures. And this quote does not state the participant’s perspective that these increased food costs by eating out led to concerns about health. A quote that illustrates this would strengthen this point.

Line 280-281: A better quote is needed to illustrate the point being made here. This might include more of the individual’s quotation that describes what the authors wrote here. It would be better to have the quote show the evidence of the idea presented.

Theme 5

Overall, this theme seems to overlap with themes 3 and 4. I think elements from this section should be merged into these other themes, as described in more detail below.

Line 289-290: Was the idea of meeting dietary requirements from the service users’ and/or service providers’ viewpoint? If not, how was this determined? From the quotation given, it sounds like the service provider’s perception. This needs to be clarified in the first sentence.

Lines 292-296: Again, how was it determined that some had difficulty meeting dietary requirements? The commentary about not being able to consume a vegetarian diet seems to overlap with theme 3 in which the participant could not eat a medically prescribed diet. Recommend moving it to theme 3.

Line 297-298: This quote does not state that the person skipped meals. Adding more to the quote that illustrates this would strengthen this idea.

Lines 299-303: This description and quotation seems like a limitation in feeding the toddler solids because of lack of cooking facilities, which was in theme 4. It seems it would fit better in theme 4.

Lines 304-310: Not having breakfast because of the limited time that breakfast is served seems like a practical barrier to healthy eating, which is theme 4.

Discussion

Lines 331-333: This last sentence about limited kitchen facilities seems out of place with the rest of the content presented in this paragraph. Recommend revising.

Lines 340-343: I think an important point is that more money was spent on eating out because of only having one meal provided and/or because of limited kitchen facilities.

Line 350-354: Authors should better integrate all information discussed related to limited kitchen facilities together (e.g., lines 331-333 and lines 340-343).

Lines 380-385: This was from the providers’ perspective, correct? This should be clarified.

Lines 401-404: For those unfamiliar with the policies around homelessness in Ireland, it could be helpful to describe in a little more detail or at least include a citation.

Conclusions

Lines 434-438 and 444-448: These conclusions seem periphery to your study findings or potential future implications of your findings and need to be revised.

Lines 439-440: Using the word linked makes this sound more like a quantitative study. Ensure that the conclusions are grounded in perceptions of those interviewed.

Author contributions: the text at the beginning was placeholder text and should be omitted so that just the authors’ names and their contributions are listed.

Suggest a table of representative quotes by themes to illustrate commonality across interviews related to the themes identified.

Author Response

Dear Dr. Lemacks,

Many thanks to you and the reviewers for their considered comments on our paper: ‘Diet Quality, Health and Wellbeing within the Irish Homeless Sector: A Qualitative Exploration’ to be considered for publication in the International Journal of Environmental Research and Public Health special Issue on "Addressing the Health Needs of Vulnerable Populations."

We would like to take this opportunity to thank you and the reviewers of our manuscript. The feedback provided was very valuable and we believe the overall quality of the manuscript has increased. Our response to the reviewer comments is highlighted below in red and has been addressed with tracked changes in the main article.

We confirm that neither the manuscript nor any parts of its content are currently under consideration or published in another journal. All authors have approved the manuscript and agree with its submission to the International Journal of Environmental Research and Public Health special Issue on "Addressing the Health Needs of Vulnerable Populations."

Thank you very much for your consideration of this manuscript.

Yours sincerely,

Divya Ravikumar

Discipline of Health Promotion

University of Galway

Reviewer 2: Line by line feedback 

Introduction

Line 25: Define OECD the first time it is used.

This has been addressed in the Introduction.

Lines 37, 45, 66: Suggest using United States instead of America.

Thank you for highlighting this. America has been changed to United States and US thereafter.

Line 48: Recommend changing the percentage to 90%.

This has been addressed.

Line 75: Recommend changing “veg” to vegetable.

This has been edited in the manuscript.

Materials and Methods

Line 110: Was a digital recorder and a mobile phone used as a security measure, to ensure interviewers were recorded? If so, that could be helpful to point this out for the reader.

Thank you for bringing this to our attention. This has been changed to clarify that a mobile phone was used as a backup measure for the digital recorder.

Lines 126-129: The “Units of study” section does not add anything meaningful to the Methods section, so recommend omitting.

Thank you. This has been removed.

Line 130: Ensure NVivo is properly cited.

We have added the appropriate citation in the manuscript.

Line 131: The phrase “audio tapes” makes it sound like you used a manual recording machine, but you digitally recorded the interviews based on line 110. Thus, more appropriate would be to call them audio files. Was the mobile phone password protected? Were the digital recorders stored in a locked filing cabinet? Adding a little more information about this would be helpful.

Thank you for highlighting this. The term ‘audio tapes’ has been changed to ‘audio files.’ The manuscript has been changed to clarify that the mobile phone used was password protected and all audio files were stored on a password-protected USB and secured in a supervisor’s locked filing cabinet in the University of Galway.

Line 132-133: I think the sentence “Data collected was only discussed anonymously with the project lead” seems out of place. I think you could omit this sentence, with the efforts used by the team to secure the data still being sufficiently described (with the additions of my comments for line 131).

This sentence has been removed.

Line 137: Ensure Microsoft OneDrive is properly cited.

Microsoft OneDrive has been cited correctly in the manuscript.

Lines 145-148: Clarify the discussion of the project with the supervisor. Did the supervisor read through all the transcripts to help decrease potential bias with only one researcher coding the data and developing themes? Also, the information seems to be written out of order – describe it in the order it occurred. For example, I am assuming the reflections were written first, then the data were coded, and then codes/themes and subthemes (however, no subthemes were identified in the Results so suggest taking out subthemes and just stating themes) were discussed with the supervisor. Some of the information in this section seems repetitive as well, so this needs to be modified.

Thank you for highlighting this. The manuscript has been changed to address your comments. In brief, data analysis, coding and identification of themes was initially carried out by one researcher. However, all codes and potential themes were discussed with the project supervisor prior to consensus on final themes.

Line 146: I am not sure what the author means by “no challenges arose during this process” – suggest omitting this statement.

This statement has been removed.

Results

Lines 151-152: How did researchers determine gender as men and women? Were they asked in the interviews, or did they complete a demographics questionnaire? This information should be described in the Methods section.

Information on participants’ gender was given to the primary researcher by service providers in hostel, emergency and day centre accommodations. This information was provided prior to commencing the qualitative interviews.

Lines 152-159: Recommend grouping all the information about service users together (length of time and residency location) and service providers together (length of time and job description).

Thank you. This has been addressed and is currently between lines 181 – 189.

Theme 1

Be consistent in clarifying if the ideas are based on service users’ or service providers’ perspectives.

Thank you. These results are based on both service user and service provider interviews and this has been clarified in the main manuscript.

Lines 167-172: These sentences seem to overlap. Recommend combining these ideas. With the quote, it could be helpful to define what “it” is referring to in brackets.

Thank you. This text has been restructured in order to avoid repetition and to enhance clarity.

Line 174: How did service users’ sense of control vary?

The original sentence was vague so it has been changed to “Although some service users within hostel accommodation were happy to have food provided for them, others felt that lack of control was an issue.”

Lines 183-193: This paragraph is somewhat hard to follow and seems contradictory in some locations. Some food donations are thrown away because of safety concerns, yet the service providers rely on supermarket food donations. Does this vary depending on the source of donations? This needs to be better clarified.

Thank you for your comment. This has been addressed by clarifying that supermarkets donate food through services such as Food Cloud while donations from the general public are thrown away.

Theme 2

Line 195: I think accommodation should be in plural form (accommodations).

This has been edited in the manuscript.

Lines 199-201: This quote is talking about the types of foods and preparation methods at hostel accommodations but from the sentence prior to this quote, it is supposed to illustrate receiving meals and snacks on site. The quote does not illustrate this.

Thank you for this comment. In order to address this, the text has been altered to ensure that it fits with the quote. The next text reads:

Participants staying in hostel accommodation had the option of receiving all meals on site and an emphasis was placed on ensuring food was freshly prepared:” “There’s always going to be veg (vegetables)….and it’s all freshly cooked. Our soups are all homemade. You know nothing comes out of a can...The lads (men) do get dessert every day” [Man, social service provider].

Lines 204-205: How are researchers defining “high quality” foods? Was this the wording given by the service users? High prevalence of sugary foods and encouraging overeating (two dinners), from a nutritional standpoint, might not be considered high quality. Perhaps the perception was that the food was tasty or highly preferred by service users?

Thank you for highlighting this. The manuscript has been altered in order to omit the term “high quality foods” as this is unclear. The sentence has been changed to: “Many service users confirmed that food was freshly prepared, plentiful and easily available within hostel accommodation”.

Line 216: Recommend adding a comma after “when sleeping rough” – also, I am assuming “sleeping rough” means the individual is a non-housed individual who is experiencing homelessness. If this is correct, it could be helpful to describe this earlier, the first time you use this term, for those who are not familiar with this term.

Thank you. The comma has been added and the term sleeping rough has been defined in the manuscript between lines 116-118.

Lines 222-225: Add quotation marks to this statement. Also, it’s a little unclear if the soup is being given for free based on the first part that says soup is for sale.

Thank you. The quotation marks have been added here and the quote has been shortened (include the quote here) in order to remove the reference to soup, as the rest of the quote illustrates the point being made.

Theme 3

Recommend moving this theme to the last one because all of the other themes describe situations that could lead to the outcomes mentioned in this theme.

Thank you for this suggestion. We have repositioned and renumbered this theme so it is now the last theme and theme 4. Considering your comments here and below, we have removed Theme 5 . However, the findings from Theme 5 have been incorporated in Theme 3 and Theme 4.

Lines 228-230: Was this mother talking about changes to cavities since being homeless? The quote is not super clear about this.

Thank you. This has been addressed by clarifying that the cavities in this service users’ child had started since their arrival into homeless services.

Line 232: It seems surprising that mothers would report such exact weight gain ranges. Did they really express ranges from 12.7 to 15.9 kg?

Both mothers reported the weight increase in their children during the interviews. They both gave the weights in stone so the text has been changed to include both stone and kilograms.

Line 241-243: Add quotation marks.

This has been addressed.

Lines 247-252: This section is somewhat hard to follow with the information provided prior to it. The sentence prior to this is talking about not having medical foods led to depression, whereas the next sentence is talking about using food to address stress or boredom, and then the last sentence talks about improved mood. Although these are related to mental and/or emotional health, they seem somewhat disconnected. Recommend revising this section for clarity. Not sure if this has been addressed properly.

Thank you for this comment. This section has been revised to make it easier to follow and information from the original theme 5 has been included here to further strengthen this section.

Line 250: What is “self-catered accommodation”? I’m not familiar with this term.

This has been addressed to clarify that ‘self-catered accommodation’ refers to accommodation with cooking and storage facilities.

Theme 4

Line 254-260: Is the lack of cooking and storage facilities in the emergency accommodation or hostel accommodation or both? It would be helpful to clarify this.

This has been addressed in order to clearly outline which accommodations have storage and cooking facilities.

Line 259-260: Add quotation marks.

This has been addressed.

Line 260-265: How does this experience in spending more money on eating out differ from the lines 254-260? It seems like this is a further example of how limited cooking facilities negatively affected food purchased and how much money was spent on food expenditures. And this quote does not state the participant’s perspective that these increased food costs by eating out led to concerns about health. A quote that illustrates this would strengthen this point.

Thank you. One of the quotes related to food expenditures has been removed as the other quote sufficiently illustrates the point. The text has also  been rephrased to highlight the impracticalities associated with the lack of cooking and storage facilities and how this negatively impacted service users. A quote has been added to strengthen this point.

Line 280-281: A better quote is needed to illustrate the point being made here. This might include more of the individual’s quotation that describes what the authors wrote here. It would be better to have the quote show the evidence of the idea presented.

Thank you. A quote that clearly illustrates the point being made has been included.

Theme 5

Overall, this theme seems to overlap with themes 3 and 4. I think elements from this section should be merged into these other themes, as described in more detail below.

Thank you for this suggestion. Theme 5 has been merged into Theme 3 & 4 in the revised manuscript.

Line 289-290: Was the idea of meeting dietary requirements from the service users’ and/or service providers’ viewpoint? If not, how was this determined? From the quotation given, it sounds like the service provider’s perception. This needs to be clarified in the first sentence.

This text has now been removed as this theme has been merged into Theme 3 & 4.

Lines 292-296: Again, how was it determined that some had difficulty meeting dietary requirements? The commentary about not being able to consume a vegetarian diet seems to overlap with theme 3 in which the participant could not eat a medically prescribed diet. Recommend moving it to theme 3.

This has been moved to what was Theme 3 (re-numbered to Theme 4 in the revised manuscript).

Line 297-298: This quote does not state that the person skipped meals. Adding more to the quote that illustrates this would strengthen this idea.

Thank you. This quote has been expanded to illustrate the point.

Lines 299-303: This description and quotation seems like a limitation in feeding the toddler solids because of lack of cooking facilities, which was in theme 4. It seems it would fit better in theme 4.

Thank you. This text has been merged into Theme 3 & 4 in the revised manuscript.

Lines 304-310: Not having breakfast because of the limited time that breakfast is served seems like a practical barrier to healthy eating, which is theme 4.

Thank you. This text has been merged into Theme 3 & 4 in the revised manuscript.

Discussion

Lines 331-333: This last sentence about limited kitchen facilities seems out of place with the rest of the content presented in this paragraph. Recommend revising.

Thank you for your suggestion. This text has been moved to later in the discussion, now line 599 - 601.

Lines 340-343: I think an important point is that more money was spent on eating out because of only having one meal provided and/or because of limited kitchen facilities.

Thank you. This has been clarified in the revised manuscript.

Line 350-354: Authors should better integrate all information discussed related to limited kitchen facilities together (e.g., lines 331-333 and lines 340-343).

This text has been integrated to ensure it is in one section, see lines 613-619.

Lines 380-385: This was from the providers’ perspective, correct? This should be clarified.

This has been clarified, see lines 620-622. This was from a service user perspective.  

Lines 401-404: For those unfamiliar with the policies around homelessness in Ireland, it could be helpful to describe in a little more detail or at least include a citation.

Thank you. Citations that reflect governmental and non-governmental organisation policy reports relating to homelessness in Ireland have been added.

Conclusions

Lines 434-438 and 444-448: These conclusions seem periphery to your study findings or potential future implications of your findings and need to be revised.

Thank you for this suggestion. This text has been rearranged to better emphasise the most important findings.

Lines 439-440: Using the word linked makes this sound more like a quantitative study. Ensure that the conclusions are grounded in perceptions of those interviewed.

Thank you for this suggestion. This text has been changed to “Food insecurity was also linked to perceived depressive symptoms and stress.”

Author contributions: the text at the beginning was placeholder text and should be omitted so that just the authors’ names and their contributions are listed.

This has been removed.

Suggest a table of representative quotes by themes to illustrate commonality across interviews related to the themes identified.

Thank you for this suggestion. A table of quotations to represent each theme has been formulated and attached as ‘Table 1.’